# Machine Learning in Tropical Cyclone Forecast Modeling: A Review

**Rui Chen [1]** **, Weimin Zhang [2,*] and Xiang Wang [2]**

[1]   College of Computer Science and Technology, National University of Defense Technology, Changsha 410073, China; chenrui16@nudt.edu.cn

[2]   College of Meteorology and oceanography, National University of Defense Technology, Changsha 410073, China; xiangwangcn@nudt.edu.cn

*   Correspondence: wmzhang104@139.com

**Abstract:** Tropical cyclones have always been a concern of meteorologists, and there are many studies regarding the axisymmetric structures, dynamic mechanisms, and forecasting techniques from the past 100 years. This research demonstrates the ongoing progress as well as the many remaining problems. Machine learning, as a means of artificial intelligence, has been certified by many researchers as being able to provide a new way to solve the bottlenecks of tropical cyclone forecasts, whether using a pure data-driven model or improving numerical models by incorporating machine learning. Through summarizing and analyzing the challenges of tropical cyclone forecasts in recent years and successful cases of machine learning methods in these aspects, this review introduces progress based on machine learning in genesis forecasts, track forecasts, intensity forecasts, extreme weather forecasts associated with tropical cyclones (such as strong winds and rainstorms, and their disastrous impacts), and storm surge forecasts, as well as in improving numerical forecast models. All of these can be regarded as both an opportunity and a challenge. The opportunity is that at present, the potential of machine learning has not been completely exploited, and a large amount of multi-source data have also not been fully utilized to improve the accuracy of tropical cyclone forecasting. The challenge is that the predictable period and stability of tropical cyclone prediction can be difficult to guarantee, because tropical cyclones are different from normal weather phenomena and oceanographic processes and they have complex dynamic mechanisms and are easily influenced by many factors.

**Keywords:** tropical cyclone; machine learning; forecasts; genesis; track; intensity; disastrous impacts; numerical forecast model

---

## 1. Introduction

Tropical cyclones (TCs) are regarded as extreme weather events, along with gales, rainstorms, and storm surges, which can cause huge losses in coastal areas worldwide. In the past century, numerous meteorologists and warning centers devoted themselves to this study and made progress in observational technology, intensification physics; interactions of the atmospheric environment, the atmospheric boundary layer and air-sea interface, the ocean responses, and forecasting techniques [1]. However, many problems with predictive skills remain, particularly with the TC genesis, intensity, and risk forecasts. Generally, the most popular tropical cyclone dynamical forecast models have a relatively low accuracy, which is mainly due to the inaccurate vortex initialization of TCs, incomplete representation of complex physical processes, and coarse resolution [2,3].

There are studies that show that insufficient representations of the air–sea energy exchange under very high wind speed conditions would hinder simulating the intensity of TCs more effectively [4].

In addition, there is also a clear view that upper ocean feedback has important effects on TCs, but few operational numerical forecast models take it into consideration, which also reduces the performance of the models [4,5]. Additionally, other methods, such as statistical models, also are unable to deal with the complex and nonlinear relationship between TC-related predictors; thus, their forecast results need to be further improved [6–10].

In order to solve these problems of traditional methods, scientists began to consider using machine learning (ML) to explore satellite, radar, in-situ data, etc. to improve the forecast skills of TCs in recent years. Machine learning algorithms, as a means of artificial intelligence (AI), can be divided into three categories according to their applications: feature selection, clustering, and regression or classification [11]. Feature selection algorithms can eliminate irrelevant attributes through attribute selection to increase the effectiveness of the tasks, and then improve the accuracy of the models. For example, a typical Tucker decomposition method can solve the spatio-temporal problems that the traditional tensor decomposition algorithm cannot deal with [12]. A clustering algorithm is one of the earliest methods used in pattern recognition and data mining and can automatically divide a sample dataset into multiple categories. This has a wide range of applications in big data analysis. Typical clustering algorithms include the finite-mixed model (FMM) [13], hierarchical clustering [14], and K-means algorithm [15].

As for classification or regression, one representative algorithm is support vector machine (SVM) for classification [16] and support vector regression (SVR) for regression [17], which can effectively deal with nonlinear problems by defining kernel functions. In addition, decision tree (DT) [18] is another typical algorithm that can mine and display the rules of classification, with high accuracy. A majority of works done with those mapping tasks are well performed with artificial neural networks (ANNs), which are considered as universal approximators for complex nonlinear mappings [19]. Since Hinton, a leading scholar of machine learning, put forward the deep neural network model in 2006, a new era of deep learning was opened. A deep neural network that contains many hidden neural layers and is excellent in feature learning, can overcome the difficulty of training through layer-by-layer initialization, and can achieve overall optimization of the network [20]. The classic networks include convolutional neural networks (CNNs) [21] and recurrent neural networks (RNNs) [22]. Compared with the traditional machine learning algorithm, deep learning (DL) has advantages in high-dimensional data, which is more suitable for complex applications. Therefore, choosing the appropriate machine learning algorithm for different data and different needs can solve practical problems more effectively.

Considering the advantages of machine learning in dealing with large-scale data, many scholars made attempts with remote sensing, meteorology, and ocean data. Applications for remote sensing include: (1) Using different deep neural networks, such as sparse autoencoder (SAE), CNN, and RNN to classify hyperspectral images and detect anomalies. (2) Interpreting synthetic aperture radar (SAR) images or high-resolution satellite images, including scene classification, object detection, image retrieval, and parameter inversion. (3) Designing fast and accurate retrieval algorithms and forward models. (4) Conducting multi-source data fusion and 3D reconstruction. The tasks of multi-source data fusion mainly include sharpening the super-resolution, fusions of feature sets and decision sets, and fusions of heterogeneous data. 3D reconstruction tasks include automatic tether point recognition and matching [23,24]. In the field of meteorology, using convolutional long short-term memory network (ConvLSTM) [25] and trajectory gate recurrent unit (TrajGRU) [26] are representative for precipitation prediction and analyzing radar images based on traditional neural networks to predict the short-term future precipitation [27], as well as the emerging use of deep neural networks for predicting the evaporation duct height of the atmosphere at the ocean surface [28]. In the marine field, machine learning focuses on the identification of mesoscale vortices or the dimensionality reduction of satellite ocean data. Several studies have shown that the networks constructed by deep learning, such as deepeddy [29], eddynet [30], and oednet [31], are of great significance for the recognition of ocean eddies.

There are still many difficulties in tropical cyclone forecasts, such as an insufficient understanding of the physical mechanisms and the complex interactions with the ocean and surrounding atmosphere environment. All of these will hinder the prediction of TC genesis, tracks, intensity, and associated disastrous weather. Machine learning proved able to provide a new way to improve the accuracy and efficiency of TC prediction. Although there may be difficulties in the current stage of machine learning in long-lead-time forecasts, in the development of a generic and interpretable ML-based TC forecast model, and in improving the numerical TC model itself, there is still a bright future for machine learning in TC forecasts because the explosive growth of multi-source data and efficient machine learning algorithms have not been well utilized. The contribution of this paper is to analyze and summarize the successful cases of machine learning in tropical cyclone forecast modeling in recent years, and then introduce them separately according to different predictive purposes. Further opportunities and challenges of machine learning in TC forecasts are described at the end, and we aim to improve the status of TC forecasts.

**This review is organized as follows:** Section 1 is the introduction, which introduces the progress and problems of TC forecasts, machine learning; the application of machine learning in remote sensing, meteorology, and ocean fields, the prospect of machine learning in TC prediction, and the organization of this paper. Section 2 summarizes the practical process of machine learning, and an overview of machine learning in TC forecasts. In Section 3, the successful cases of machine learning in TC forecasts modeling in recent years are divided into five categories, and a detailed summary of them is described. Section 4 is our reflections on the above problems and progress, indicating the opportunities and challenges for machine learning in future tropical cyclone forecasting. Section 5 is our conclusions.

## 2. Machine Learning

### 2.1. A Brief Introduction to Machine Learning

Machine learning is a series of computer programs, and its core task is to build mathematical models using statistics to make inferences from samples. Given a model that defines certain parameters, learning is the execution of a computer program that uses training data or experiences to optimize the parameters of the model. The model can predict the future state, or describe the knowledge from the data, or both. The practical process of machine learning methods (see Figure 1) can be summarized as follows: (1) Define a problem to an unknown mapping $f$ and set a hypothetical set $H$ of the solving model. (2) Collect and organize a training set $D$ with a finite set. (3) Specify the loss function for the model. (4) Select the learning algorithm $A$. (5) Obtain the parameters that make the loss function fetch the pole hour and choose them as the optimal parameters of the model. (6) Save this model $g$ with the optimal parameters, and use it to make predictions and analysis of new data.

Machine learning algorithms can also be divided into several categories according to the learning tasks, such as prediction, feature selection, and dimensionality reduction. As this review focuses on TC forecast modeling, only predictive algorithms will be described here. Generally, if the goal of the model is to predict discrete values, this kind of learning task is called "classification"; if it is to predict continuous values instead, this learning task is called "regression". In addition, learning tasks can also be broadly classified into "supervised learning" and "unsupervised learning" depending on whether the training data are labeled or not, with classification and regression representing the former and clustering representing the latter. The prediction task intends to establish a mapping $f$ from the input space $X$ to the output space $Y$, $f: X \rightarrow Y$, and $f$ depends on a vector of nonlinear (in general) parameters, $w: y = f(x, w)$. The parameters w are obtained in the process of training, which, for the classification or regression/mapping problem, is an optimization of the performance criterion (e.g., a minimization of the mean square error). Of course, a machine learning algorithm itself may have additional parameters (hyperparameters), such as the number of hidden neurons and learning rates for neural networks. The selection of hyperparameters also plays a crucial role in training an appropriate machine learning-based forecast model. This section mainly referred to [32–34].

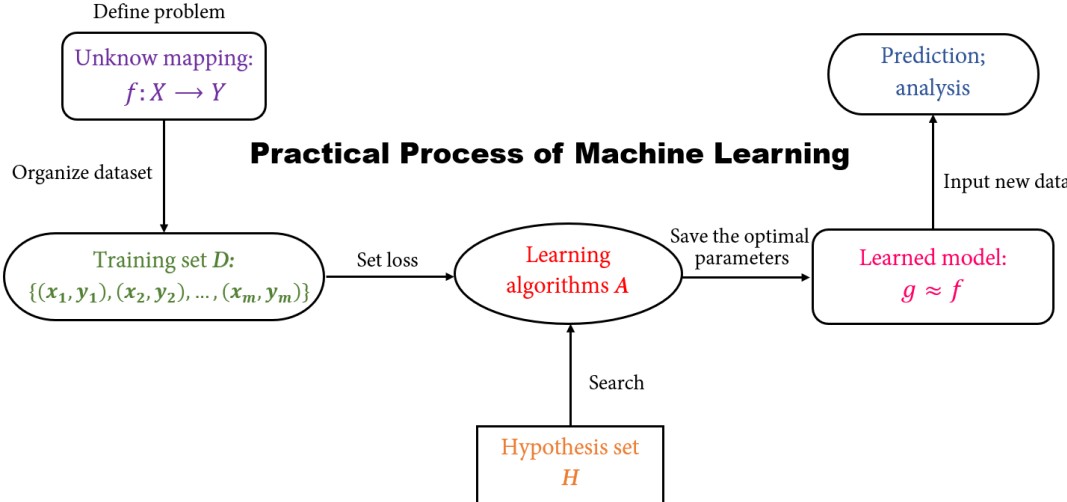

**Figure 1.** This is the practical process of machine learning.

## 2.2. An Overview of Machine Learning in TC Forecasts

TC forecasts focus on the prediction of the central location and intensity of TCs, as well as on the effects of catastrophic weather when they make landfall or come close to shore, and the forecast techniques are mainly empirical forecasting, statistical forecasting, and numerical forecasting models. In recent years, path prediction has made huge progress, apart from some abnormal paths, normal path prediction can achieve relatively accurate results. Existing techniques for predicting the genesis and intensity of TCs are still very limited, and the reasons include (1) TC genesis and intensity are not as well defined as TC location, and (2) the physical processes involved are more complex and difficult to describe precisely by statistical models or dynamic equations. In addition, the forecasts of high winds, rainstorms, and storm surges in TC-affected areas highly rely on accurate trajectories and intensity forecasts, which makes their current forecasts even more worrisome. For machine learning, as mentioned in Section 2.1, the major application is to perform predictions. This type of method discovering rules from data and unconstrained by physics is particularly suitable for solving problems where the physical mechanisms are unclear, such as TC changes. Therefore, if there are sufficient historical TC samples and a large amount of relevant meteorological and oceanic data, machine learning is expected to accurately predict TC events. Of course, the explosive growth of satellite data, observational data, and re-analysis data offers tremendous opportunities for machine learning in TC forecasts.

As shown in Figure 2, the applications of machine learning in TC forecasts can be divided into five aspects. Regarding TC genesis forecasts, the final goals are to generate probabilistic forecasts of a fixed region in real time and quantitative forecasts in the time and place of cyclonegenesis, so as to better monitor the tropical ocean. However, at this stage, machine learning is only capable of predicting whether the precursors can evolve into TCs, and the seasonal frequency of TC genesis in each area, which corresponds to a classification task and regression task in machine learning, respectively. Thus, researchers primarily use several typical algorithms, including DT, logistic regression (LR), SVM, and ensemble algorithms, like AdaBoost and random forest (RF), for TC genesis prediction. These ensemble algorithms are theoretically better than a single algorithm; however, they still need to be judged on a case-by-case basis. Additionally, deep learning algorithms, such as multi-layer perceptrons (MLPs) and CNNs, which can better fit complex functions and process image data, also play an important role in improving TC genesis forecasting techniques.

For TC track forecasts, machine learning-based models are commonly derived from statistical learning methods, i.e., using the characteristics of the TC itself and the associated meteorological and oceanic variables to predict the position of TCs, and this is considered nonlinear mapping. MLP and

RNN, as the researchers' first choice, were also proven to be more effective than traditional methods. In addition, TCs change in both time and space; therefore, the spatio-temporal models, like ConvLSTM, have also achieved good results in this prediction task. In addition to those classic ideas, there is also the attempt of using generative adversarial networks (GANs) to generate predicted TC cloud images and then locate the TC center on them, or the use of deep belief networks (DBNs) and clustering algorithms like FMM to search for historically similar paths for forecasting. Each of these methods achieved good experimental results in their papers, but it remains to be seen whether better results can be achieved in operational forecasting than with existing techniques. DT, as an algorithm capable of mining rules, can be well used to mine the predictive factors and rules for TC landfall and recurvature, thus laying the foundation for track forecast modeling in the future.

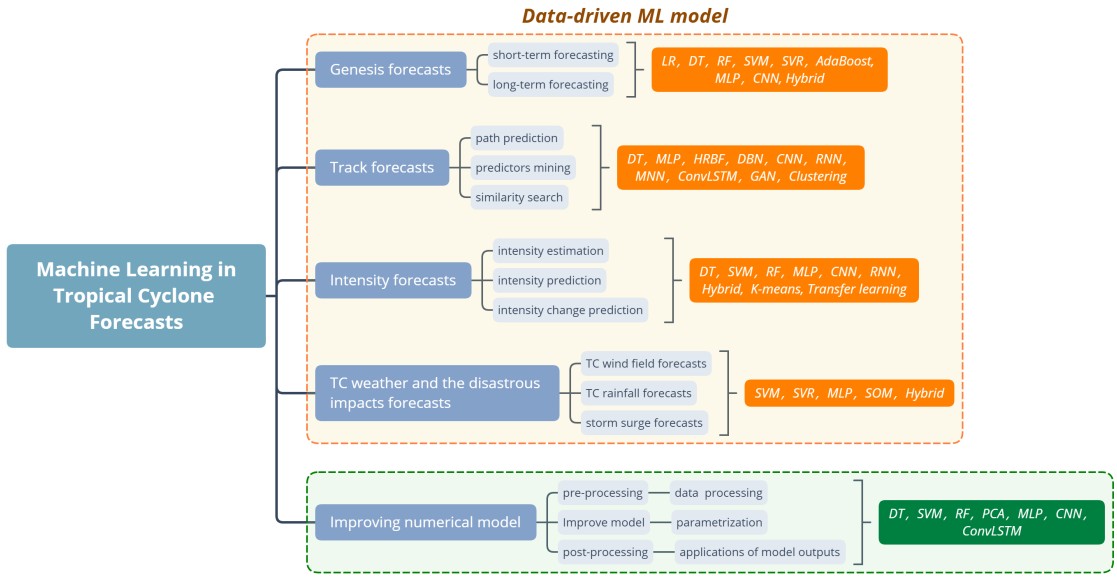

**Figure 2.** An organization chart of cases involving machine learning in tropical cyclone forecasts. The abbreviations used in this figure are as follows: logistic regression (LR), decision tree (DT), random forest (RF), support vector machine (SVM), support vector regression (SVR), multi-layer perceptron (MLP), convolutional neural network (CNN), generative adversarial network (GAN), recurrent neural network (RNN), hybrid radial basis function network (HRBF), self-organizing map (SOM), principal component analysis (PCA), convolutional long short-term memory network (ConvLSTM).

For intensity forecasts, satellite data have been an important data source, due to the difficulty of obtaining data from in-situ observations. Conventional Dvorak technology can detect TCs from satellite cloud images and then determine the central position and intensity of TCs; however, this method is not sufficiently objective and accurate. Based on a similar idea, many researchers employed CNN to replace Dvorak to better estimate the intensity of TCs. The typical method of directly predicting the intensity is to take the whole path of a TC as a sequence and to use MLP or RNN to predict the intensity of the next moments based on the intensity of the current moment. Certain studies incorporated meteorological and oceanic variables into their predictive model and then discovered the spatio-temporal information in the data using a hybrid network of a CNN and long short-term memory (LSTM), which has been shown to further improve the forecast results.

Currently, these methods are only effective for short-term forecasting in local areas and there is a long way to go before they can be applied to operational forecasts. The problem of rapid intensification (RI) is also one of the key factors hindering intensity prediction. Thus, if rapidly intensified TCs can be well predicted to build a specific RI-TC model, the intensity prediction will certainly be effectively improved. Currently, RI prediction is regarded as a classification problem in machine

learning (i.e., to predict whether RI will occur), or as a regression problem to predict the probability of RI occurrence. The main algorithms used include DT, SVM, and RNN.

In terms of TC weather and its disastrous impact forecasts, machine learning methods for predicting rainfall and storm surges are relatively similar. Generally, the characteristics of TCs, relevant precipitation, and tide information are used to predict rainfall or storm surge height. MLP and SVR with these predictors can provide significant improvements compared to statistical models that do not take into account nonlinear relationships and dynamic models whose physical mechanisms are not clear. Other hybrid networks that add a self-organizing maps (SOM) algorithm are also based on the same idea; however, they make the network model more efficient by further processing the data. For wind field forecasts, CNN is most commonly used to simulate the wind field in the core or boundary layer, and SVR can also be used to predict the wind field near the ocean surface. However, at this stage, it is not possible to predict the entire TC wind field well.

Apart from the pure data-driven machine learning methods mentioned above for TC genesis, tracks, intensity, and disastrous weather impact forecasts, there is another way to improve forecast results by developing a physics-based machine learning model. Although the existing techniques are not sufficient to comprehensively improve numerical forecast models through machine learning, there have been some successful cases regarding this topic. Here, we will briefly divide them into three categories.

The first is pre-processing, which includes the quality control of data used to construct the initial field of the model. For example, SVM can be used to determine if there is a TC region in the data, either by eliminating it, or by performing special processing to improve the quality of the data. The second is the improvement of the model itself, including model error correction and an improved parameterization scheme. Existing studies used only MLP and CNN to quantify sea surface temperature cooling (SSTC) induced by TCs in Weather Research and Forecasting (WRF) to improve the numerical forecasts of TC intensity, or they used improvements in parameterizing the TC wind field based on RF and principal component analysis (PCA). The third category is post-processing, which includes model output corrections and applying for numerical products. However, only applications of numerical model products for predicting TC genesis, paths, intensity, rapid intensification, etc., have been determined currently, and, to the best of our knowledge, no studies on revising the TC numerical forecast results have been found. Therefore, there are only some preliminary studies for machine learning in this aspect, and there is still room for improvement. The details of all cases of machine learning in TC forecasts will be expanded in the next section.

## 3. Applications

### 3.1. Genesis Forecasts

Tropical cyclone genesis, also called TC formation, is a significant problem in TC research and forecasting. In general, TC genesis consists of two stages: the first is the transformation from tropical disturbance to tropical depression, and the second is the development from tropical depression to tropical storm, which may be governed by physical mechanisms, such as convective instability of the second kind (CISK) or wind-induced surface heat exchange (WHISE) [35,36]. In previous studies, tropical cyclogenesis was considered highly dependent on large-scale environmental factors. Thus, Gary first identified six TC-related environmental factors: the Coriolis parameter, the low-level relative vorticity, the shear of the horizontal wind through the depth of the troposphere, the relative humidity of the middle troposphere, the ocean thermal energy, and the difference between the equivalent potential temperatures at the surface and at 500 hPa [37]. These factors have evolved in later studies and have been used for different predictive purposes, including short-term genesis prediction and long-term (seasonal) genesis frequency prediction.

Traditional forecasting techniques primarily include numerical and statistical models. Global numerical models, in particular the European Centre for Medium-Range Weather Forecasts

(ECMWF), the Global Forecast System (GFS), the UK Meteorological Forecast Model (UKMET), and their ensembles, are the main tools for predicting TC genesis, assessing the development of the disturbance itself, and studying the environmental factors that contribute to tropical storms [38]. Although this predictive method is based on physical interpretations, it has limitations on the poor understanding of TC genesis and huge computational costs. For statistical models, including linear discriminative analysis (LDA) and naïve Bayes (NB), they build the linear relationship between the probability of TC genesis and large-scale environmental predictors [39]. However, linear functions are not sufficient to accurately describe the complexity between them, and so it is difficult to improve the prediction results further. Therefore, machine learning was used for genesis forecasts, and all cases are shown in Table 1.

**Table 1.** Machine learning in tropical cyclone (TC) genesis forecasts.

| Tasks | Algorithms | Main Idea | Reference |
|-------|-----------|-----------|-----------|
| Short-term forecasting | LR | Select the optimal predictors and modeling for genesis forecasts | Wijnands, J.S. (2016). [40] |
| | DT | Predict future tropical cyclogenesis based on tropical perturbations | Zhang, W. (2015). [41] |
| | | Detect the causes of TCs by using predictors from satellite data | Park, M.S. (2016). [42] |
| | RF | Predict the development of MCSs | Ahijevych, D. (2016). [43] |
| | AdaBoost | Determine whether MCSs will evolve into TCs | Zhang, T. (2019). [44] |
| | SVM | Predict TC formation from satellite image data | Kim, M. (2019). [45] |
| | CNN | Detect TCs and their precursors based on the simulation of numerical models | Matsuoka, D. (2018). [46] |
| Long-term forecasting | SVR | Generate forecasts of the TC activity for an upcoming season | Richman, M.B. (2012). [47] |
| | | Reduce TC seasonal prediction errors | Richman, M.B. (2017). [48] |
| | | Improve the accuracy of seasonal TC predictions | Wijnands, J.S. (2014). [49] |
| | MLP | Provide seasonal prediction of TC activity | Nath, S. (2016). [50] |
| | SOM, FNN | Define GPI for a ensemble of global climate models | Yip, Z.K. (2012). [51] |

### 3.1.1. Short-Term Forecasting

A machine learning-based predictive model is a data-driven model that can be divorced from existing unspecified physical mechanisms and is highly advantageous in capturing non-linear relationships between predictors and predicted objects. In recent literature, short-term genesis forecasting was often defined as a classification problem of machine learning, for predicting whether TC precursors will evolve into TCs. There is a study [40] that presented how to select the optimal variables for short-term forecasting (up to 72 h) of TC genesis using the Peter–Clark algorithm. Predictive modeling with LR confirmed the superior performance of the top-ranked variables. The presented variable ranking can also be used for the creation of genesis indices or predictive models in the future.

The authors of [44] attempted to determine whether mesoscale convective systems (MCS) evolved into tropical cyclones at different lead times using nonlinear classifiers (decision tree (DT), K-nearest neighbor (KNN), MLP, qualitative data analysis (QDA), SVM), or nonlinear ensemble classifiers (AdaBoost and RF)). The results showed that AdaBoost was the most effective algorithm with a forecast accuracy of 97.2% (F1-score) for tropical cyclone genesis compared to conventional linear statistical models using the environmental predictors associated with MCSs/TCs over a prediction time of 6 h. Robustness was also assured when the lead time was extended to 12, 24, or even 48 h.

Similarly, Ahijevych et al. [43] used RF to produce 2 h predictions of the development of MCSs and found that RF had a significant ability in detecting MSCs. For the 550 observed MSCs, 99% accuracy was achieved (within 50 km). In addition, the DT algorithm in [41] was used to predict future tropical cyclonegenesis based on tropical perturbations defined by vorticity fields from the Navy Global

Atmospheric Prediction System (NOGAPS). The algorithm derived six classification rules based on environmental predictors and the prediction accuracy of the test data set was 84.6%. In addition to re-analysis data or simulation data, the data source for the predictors can be satellite data. For example, Park et al. [42] qualified eight predictors from the ocean surface wind and precipitation measured by WindSat satellites, and then used these predictors to design a DT-based detection model to detect the cause of TCs. The validation results showed that the model had a positive detection rate of about 95.3% and a false positive rate of 28.5%, which is comparable to the previously available objective methods based on cloud pattern recognition.

DT, RF, and SVM were also compared for their prediction skills (see [45]). Thereby, SVM was the most effective machine learning algorithm, performing better in prediction skills compared to the other algorithms, with hit rates ranging from 94% to 96%, which was significantly higher than the LDA performance (77%). Deep learning can be used to solve this problem (e.g., [46]). That paper employed CNNs to automatically capture features of precursors or TCs, rather than constructing predictors associated with precursors of TCs as features. The performance of the CNNs was studied in different basins, seasons, and lead times, and the best result was 91.2% detected precursor formation at a lead time of two days in the Western North Pacific.

### 3.1.2. Long-Term Forecasting

Long-term genesis forecasts are aimed at predicting the numbers of TCs in vulnerable areas for the upcoming TC season, which facilitates the preparation of emergency services months in advance. In order to obtain better seasonal predictions of the TC frequency, machine learning algorithms are also suitable, as this can be considered a regression problem. Richman et al. [47] used support vector regression (SVR) to predict the number of seasonal TCs. In their experiment, SVR had the same predictors as the multiple linear regression (MLR) in the previous study, but with a 40.1% improvement in the results compared to MLR. To further improve the result, the authors also incorporated quasi-biennial oscillations (QBO) into the SVR, which resulted in a significant 121% improvement in the predicted results of SVR, while the mean absolute error (MAE) (0.97) and root mean square error (RMSE) (1.19) improved by 31.2% and 29.2%, respectively.

There are many similar studies ([48–51]) where the authors employed SVR and MLP for the prediction of TC activity, along with designing different potential predictors. Defining and calculating the genesis potential index (GPI) is another method to predict the rate of genesis per unit area per unit time apart from forecasting the numbers of seasonal TC genesis directly. The GPI is a single-valued function that represents the empirical relationship between the relevant predictors of storm formation and the potential of TC genesis. Considering that GPI is non-linear and that the environmental variables are not independent, a combination of SOM and a feed forward neural network was used in [51] to study changes in the probability of tropical cyclone genesis in the future and to discover the main variables for the predicted changes. This method measured the GPI based on the output of global climate models (GCMs) and was shown to produce more objective analyses and reliable results than TC-like vortices simulated by GCMs.

### 3.2. Track Forecasts

TC track forecasting is usually based on common but comparatively accurate understandings of TC movement, which is affected by many complex factors, such as large-scale weather patterns, the sea surface and atmosphere temperatures, features of land topography, TC structure, and TC intensity. At present, a normal path can be well predicted, but there are still difficulties in dealing with abnormal path prediction problems, such as sudden changes in the move speed, recurvature, and even stagnation. As the occurrence of abnormal paths is an extremely complicated dynamic problem, it is difficult to precisely describe in existing methods. Thus, it remains a key point in operational forecasts [1].

The traditional methods for track forecasts can be divided into six categories: weather maps analysis, diagnostic analysis, mathematical statistics, dynamic model, statistical-dynamic models,

and climate extrapolation. Although dynamical models, which are generally known as numerical forecast models, are the primary method, mathematical statistics as a supplementary method have played an increasingly important role in this aspect. The mathematical statistics model considers that the atmospheric motion is a random process with laws that cannot be comprehensively described with a set of equations; however, the statistical characteristics or laws can be determined to predict the path.

Additionally, the statistical-dynamic model, which is one of the modern forecast techniques for TC paths, can also reference future machine learning methods. This includes (1) Selecting dynamic factors, such as the vorticity, divergence, vorticity, advection, etc. as inputs into the statistical model. (2) Selecting the results of numerical forecasts to be input into the statistical model as predictors. (3) Statistically testing the results of the numerical forecasts, and then correcting the statistical deviations into the numerical model. (4) Establishing a numerical model combining dynamics and statistics, which can not only retain the basic dynamical mechanism described by the physical equations, but also deal with the uncertain quantities in the dynamical process by statistical means [52,53]. In recent years, to produce a more effective and quick result for TC path forecasts, many researchers attempted to apply machine learning to build new predictive models (see Table 2), and they mainly focused on the improvement of prediction techniques and the selection of predictors.

**Table 2.** Machine learning in TC track forecasts.

| Tasks | Algorithms | Main Idea | Reference |
|---|---|---|---|
| Path prediction | HRBF | Develop an automatic and integrated TC identification and track mining system | Lee, R.T. (2000). [54] |
| | MLP | Predict cyclone tracks based on MLP with BP algorithms | Ali, M.M. (2007). [55] Wang, Y. (2011). [56] |
| | RNN | Propose a sparse RNN with flexible topology for trajectory prediction | Moradi, K.M. (2016). [57] |
| | | Propose a fully connected RNN to predict the trajectory of TCs | Alemany, S. (2019). [58] |
| | MNN | Develop a predictive model that preserves spatial information from the cyclone tracks | Wang, Y. (2018). [59] |
| | GAN | Predict TC tracks using GAN with satellite images and meteorological data | Rüttgers, M. (2018,2019). [60,61] |
| | ConvLSTM | Propose a ConvLSTM-based spatio-temporal model to track and predict TC trajectories | Kim, S. (2019). [62] |
| | CNN | Design a model fusing multi-source data to predict TC tracks | Giffard, R.S. (2020). [63] |
| Predictors mining | DT | Discover predictors and rules governing TC landfall and recurvature | Zhang; Geng (2013, 2016). [64–66] |
| Similarity search | DBN | Find the similar TCs in history and reference this data to improve TC forecasting | Wang, Y. (2018). [67] |
| | Clustering | Apply K-means, fuzzy c-mean, and SOM for clustering TC tracks | Jinhua; Kim (2011, 2016). [68–70] |
| | | Study TC properties and large-scale factors | Camargo; Ramsay; Zhang; Geng (2007, 2008, 2012, 2013, 2016) [66,71–75] |

### 3.2.1. Path Prediction

Lee [54] proposed a TC recognition and trajectory mining system 20 years ago, consisting of two main modules based on neural networks: the first is a TC pattern recognition system using a neural oscillatory elasticity map matching model (NOEGM); and the second is a TC trajectory mining system using a hybrid radial basis function (HRBF) network with a time difference and structure learning (TDSL) algorithm. In the experiments of TC pattern recognition of satellite images, the correct segmentation rate reached 98% and the correct classification rate exceeded 97%. In addition, the hybrid RBF networks also achieved results of more than 86% in the mining tests of TC trajectory. The proposed

RBF network improved the prediction error by 30% and 18% compared to the one-way-interactive tropical cyclone model (OTCM) used by the Joint Typhoon Warning Center (JTWC) in Guam and the track forecast system (TFS) used by Central Weather Bureau in Taiwan.

After that, the authors of [55] used MLP to predict the next 24-h latitude and longitude of the cyclone by inputting the observations of the past 12 h. The results showed that the RMSE between the predicted value and the actual latitude (longitude) was 1.01 (1.16) degrees and the correlation coefficient of the predicted value was 0.98 (0.99). Compared to the Climatology and Persistence (CLIPER), Florida State University based Limited Area Model (LAM), and Quasi Lagrangian Model (QLM), MLP achieved a more accurate effect. However, they found that the prediction accuracy decreased beyond 24 h, proving that MLP still has significant limitations in forecasting. There is a similar study using MLP with BP algorithms for forecasting western Pacific tropical cyclone tracks, which have similar performance [56].

In order to perform longer predictions, some researchers considered TC motion as a time series, and RNN was used for coping with this problem by adding the previous information to the current input to provide more effective prediction results [57,58]. For example, Ref. [58] used RNN to reduce the truncation errors in statistical-dynamical models, and this algorithm can not only capture the features of complex and nonlinear weather data but also give a 120-h lead time of track forecasts. Compared to the techniques currently used in National Hurricane Center (NHC) to predict TC trajectories, the prediction accuracy is higher. Zhang et al. [59] developed a matrix neural network model (MNN) that preserves spatial information from the cyclone tracks, and the results showed that the proposed method was more effective when compared to other methods (gated recurrent unit (GRU), LSTM, MLP, and RNN) given the chaotic nature of cyclone tracks.

With the development of deep learning algorithms, many specific network structures appear for specific problems, such as GAN. GAN is also used to deal with the satellite image data to achieve a higher TC track forecast accuracy. Considering that GAN can use the images from the past to automatically generate an image that shows the future TC center and the cloud structure, it is extremely suitable to forecast the trend of TCs [61]. Errors between predicted and real TC centers are measured quantitatively in kilometers, and an averaged error of 95.6 km was achieved for 10 tested TCs. They also discovered that adding physically meaningful meteorological data to satellite images could improve the sharpness of the predicted images; therefore, GAN could also predict certain specialized TC movements, such as rotary motion. After this operation, the overall error was 67.2 km, compared to 95.6 km when only observational satellite data were used as input [60].

TC's motion occurs on both spatial and temporal scales, and ignoring either of these can reduce the accuracy of the predictive effect. Thus, researchers attempted to use products of numerical models as input to construct spatio-temporal models based on ConvLSTM, such as the Deep-Hurricane-Tracker for TC tracks [62]. Experiments showed that the model performed significantly better than the existing baseline. In addition, while there are already many classic networks, building fusion neural networks can be another way to improve the applications of machine learning methods. In [63], the authors presented a neural network model that incorporated past trajectory data, reanalyzed atmospheric images (three-dimensional fields of wind and pressure), and used a moving frame that followed the center of the storm for 24 h tracking forecasts. The most obvious advantage of this method is that it can provide storm forecasts in seconds, which is an important advantage over traditional forecasts in real time.

### 3.2.2. Predictors Mining and Similarity Search

In addition to improving forecasting techniques through machine learning algorithms, there is another application in trajectory prediction, which is to explore valuable predictors for building statistical or machine learning-based forecasts models. According to [64,65], the DT method was used to discover the contributors and rules of TC recurvature and landfall. The potential parameters that affect TC recurvature are usually divided into three categories: large-scale circulation,

circulations surrounding TCs, and variables characterizing TCs. Through using parameters measuring large-scale circulation patterns and characterizing TCs and applying a DT algorithm (C4.5) to classify recurving and straight-moving TCs, researchers derived rules that can be explained by existing theories and that are supported by various empirical findings on TC recurvatures. Similar findings have been made in experiments with landfalling TCs.

Considering the vast number of historical TC cases, researchers designed a suitable search algorithm to help forecasters find similar historical TC cases to achieve the final forecast goals. There is a historical TC similarity search algorithm [67] that employed deep learning approaches based on 500-hPa weather patterns, and this algorithm can automatically extract weather features using DBN. The results showed that the modified similarity TC trajectory method improved the prediction results (at 85% confidence) when the lead time was 54, 60, or 66 h.

Cluster analysis also could be potentially useful in forecasts. Once a TC is identified as belonging to one of the clusters, the historical information about certain paths can be used as a guidance for forecasts. K-means, fuzzy c-means, and SOM have been employed for clustering TC tracks [68–70]. Another commonly used algorithm is the FMM-based method. This objective method was developed in [71–74] to classify the typhoon tracks based not only on a few points of the path but also on the shape and location of the path, which has been used to study the general properties of TCs and how the large-scale circulation, El Niño-Southern Oscillation (ENSO), and Madden–Julian Oscillation (MJO) affect each cluster. FMM was also used for clustering post-landfall tracks of TCs making landfall over China, to unravel the patterns and structures of post-landfall TC tracks [75].

The authors of [66] combined FMM with classification and regression tree (CART). The tracks of TCs landfalling over the Chinese coast were classified into three clusters through an FMM, and several climate indices were analyzed using the CART algorithm for the three clusters. The results showed that the FMM algorithm was effective for the track classification of TCs landing over China, and the CART algorithm, which can be explained and understood easily, showed high prediction accuracy.

### 3.3. Intensity Forecasts

Unlike track forecasting, which has made clear progress in recent years, intensity forecasts remain a challenging topic in operational forecasting. TC intensity is usually defined as the maximum wind speed or minimum sea level pressure at the center of a TC, but specific definitions of TC intensity vary in different oceanic zones, and there is no standard definition to date [76]. In fact, for the definition of the TC center, the maximum wind speed or minimum sea level pressure are key factors for achieving realistic and effective forecast results. Although existing forecasts are based on those uncertain definitions of TC intensity, this does not limit the development of intensity forecast techniques. This is likely because intensity is simply a metric to quantify the severity of a TC, rather than the final goal of this type of extreme event forecasts.

From previous observations and studies, in most cases, the change in intensity over a period of time is usually slow and steady; however, there are some anomalies in which the TC intensity changes rapidly. These anomalous TCs are the difficulty of the current forecasts [77]. For the reasons why intensity predictions lag significantly behind track predictions, most researchers believe that the physical processes of intensity change are so complex that humans understand very little about them [78,79]. There are many factors, such as air–sea interactions and large-scale atmosphere environments, that influence TC intensity change and are hard to explain, so neither numerical nor statistical-dynamic models can give accurate intensity predictions [80,81].

However, how to improve the accuracy of intensity predictions to provide more accurate disaster warnings is an issue that countries around the world urgently need to address [82]. As a result, researchers and forecasters are beginning to make intensity predictions directly or indirectly through machine learning. The objectives for all cases were focused on three main categories: predict whether intensity changes rapidly or not, predict future intensity. In the description of the success cases that

follow (see Table 3), it is not difficult to find that machine learning algorithms perform well on intensity prediction problems.

**Table 3.** Machine learning in TC intensity forecasts.

| Tasks | Algorithms | Main Idea | Reference |
|---|---|---|---|
| Intensity estimation | DT | Estimate intensity using microwave image data | Bankert, R.L. (2003). [83] |
| | SVM | Propose a machine learning framework in labelling TC intensity levels | Chen, Z. (2018). [84] |
| | | Design a network architecture for categorizing TCs based on intensity | Pradhan, R. (2018). [85] |
| | | Explore the possibilities of estimating TC intensity from satellite images | Wimmers, A. (2019). [86] |
| | CNN | Estimate TC intensity as a regression task | Chen, B.F. (2019). [87] |
| | | Employ 2D-CNN and 3D-CNN to analyze the relationship between satellite images and TC intensity | Lee, J. (2020). [88] |
| Intensity prediction | MLP | Predict TC intensity values directly | Jong-Jin, B. (2000). [89] |
| | | Compare different network-based models to identify the best intensity forecasts model | Chaudhuri, S. (2013). [90] |
| | RNN | Design a pure data-driven intensity prediction model | Pan, B. (2019). [91] |
| | CNN-LSTM | Design a spatio-temporal model based on a hybrid network of 2D-CNN, 3D-CNN, and LSTM | Chen, R. (2019). [92] |
| | Transfer learning | Develop a robust prediction model with transfer learning and stacking | Deo, R.V. (2017). [93] |
| Intensity change prediction | EA, PSO | Apply EA or PSO to predict whether TCs will intensify or weaken | Geng, H. (2015,2017). [94,95] |
| | DT | Validate RI related predictors and predict intensity change | Zhang; Gao (2013, 2016). [96,97] |
| | RNN | Employ RNN with cooperative coevolution for RI prediction. | Zhang, W. (2013). [98] |
| | SVM | Apply ML techniques to classify storms as RI or non-RI | Mercer, A. (2015). [99] |
| | SVM, ANN, RF | Quantify the RI predictability using an ensemble of AI methods | Mercer, A. (2017). [100] |
| | K-means | Explore TC-troughs configurations that are favorable for RI | Fischer, M.S. (2019). [101] |

### 3.3.1. Intensity Estimation

Real-time intensity estimation using satellite data is usually considered as an effective way to monitor TCs, and has been used by many researchers. The development of estimation algorithms is also important for improving forecasting techniques, as these can identify the features of satellite cloud images to provide an estimation of intensity based on the features of the current images, and even make predictions based on future cloud images generated by the algorithm. According to a previous study, the DT method was applied for the first time for TC intensity estimation using special sensor microwave/image (SSM/I) data, with an overall RMSE of 18.1 kt for 98 independent images and a MAE of 14.3 kt [83]. The authors of [84] considered intensity estimation as a classification problem, using the cyclone intensity levels as the class labels. Thus, they proposed a machine learning framework consisting of three main parts: usable band determination, band-wise classification, and fusion. Experiments using MLR, SVM, and MLP sequentially as band classifiers showed that SVM could obtain the highest accuracy for TC intensity category identification.

The primary data source for intensity estimation is satellite imagery data. Deep learning methods, such as CNN, have their own unique advantages when it comes to imagery problems. As a result, researchers began using CNN for more efficient intensity estimation operations in subsequent studies. For example, Pradhan et al. [85] designed a CNN-based architecture for categorizing TCs according

to their intensity. The model obtained better accuracy and lower RMSE than the state-of-the-art techniques of intensity estimation using satellite imagery, and it could visualize the features at different layers and their deconstruction to understand the learning process. From this study, another model, "DeepMicroNet", emerged with unique features, including making probabilistic outputs, being able to revise partial scans, and being able to fix an inaccurate TC center [86]. The RMSE of the model was 14.3 kt (1 kt $\approx$ 0.514 m/s) compared to the best track records, while this improved to an RMSE of 10.6 kt when compared to the aircraft reconnaissance-aided best track dataset.

A similar study showed an RMSE of 8.39 kt for intensity estimation according to the best track records, and the RMSE for 482 samples analyzed with reconnaissance observations reached 8.79 kt [87]. In the latest study, the authors of [88] used two-dimensional CNN (2D-CNN) and three-dimensional CNN (3D-CNN) to analyze the relationship between multispectral geostationary satellite images and TC intensity. Their optimal model produced an RMSE of 8.32 kt, which was better (35%) than the existing CNN-based model with a single channel image. In addition, the features of the TC intensity based on multispectral satellite images can also be analyzed by heat maps, which is one of the visualization methods of CNN.

### 3.3.2. Intensity Prediction

As for predicting intensity directly, as early as the beginning of the 21st century, there were researchers who used MLP with BP algorithms to build 12–72 h intensity prediction models in the Western North Pacific (WNP) [89]. The predictive factors of the neural network were selected on the basis of the predictive factors of the MLR, but the mean error of this forecast model was 7–16% smaller than that of the MLR model. Subsequently, a similar study adopted MLP to predict TC intensity, with different predictors in [89], and the new model is considered as an alternative to traditional operational forecasting models [90]. Similar to path prediction, intensity prediction can be seen as a time series prediction. Therefore, RNN was used for building prediction models with lead times at 24 and 48 h. Compared to dynamical models, such as the Japan Meteorological Agency-Global Spectral Model (JMA-GSM) and statistical model, the models performed better in reducing the final error, with an average error of 5.1 m/s for 24 h and 6.7 m/s for 48 h [91].

The prediction of intensity is not only a temporal problem but also a spatial problem. Therefore, Chen et al. [92] believed that it was necessary to construct a spatio-temporal model for intensity prediction that effectively exploits the relationship between multiple factors affecting the TC intensity to obtain better prediction results. They built a hybrid CNN-LSTM model based on classical deep neural networks like CNN and LSTM, and this model proved to have less error (7.9 kt in West Pacific (WP)) than some existing numerical models, statistical models, and traditional machine learning methods. In addition, transfer learning can be used to complement the target dataset by incorporating knowledge from the dataset. Thus, the authors of [93] developed an effective strategy to evaluate the relationship between different types of cyclones through transfer learning and traditional learning methods, and to then predict the intensity more consistently.

### 3.3.3. Intensity Change Prediction

From previous studies, the purpose of intensity change prediction is not to directly predict intensity values, but rather to predict whether the TC will intensify or weaken over time. Due to some difficulties in predicting intensity directly, the prediction of change in TC intensity, especially with RI, is also an important complement to direct intensity forecasting. Evolutionary algorithms (EAs), hierarchical particle swarm optimization (PSO) algorithms, and DT methods (e.g., CART or C4.5) have been used to forecast TC intensity changes [94–96]. A large part of the reason for the inaccuracy of intensity forecasts at this stage is that it is difficult to make accurate predictions on issues, such as RI. Thus, scholars who are attempting to improve TC forecasts have focused on RI.

However, the definition of RI suffers from the same problem as the definition of intensity, lacking an international standard [102]. For example, NHC defines RI as a 30 kt increase in the

sustained peak wind speed over a 24 h period, while the National Weather Service defines RI as a 42 mb or greater decrease in central pressure over a 24 h period. Ignoring these imprecise definitions, researchers tend to focus on predicting whether RI will occur and mining the key factors of predicting RI. The authors of [97] improved RI prediction by incorporating an ocean coupling potential intensity index (OC_PI) in DT. The DT with the OC_PI showed a cross-validation accuracy of 83.5% and an independent verification accuracy of 89.6%, which outperformed the DT excluding the OC_PI with a corresponding accuracy of 83.2% and 83.9%.

The authors in [98] defined RI prediction as a classification problem, and used RNN to predict it. The results showed that the neural network performed better for most cases except for the extreme case where the intensity varied more than 30 kt within 24 h. The authors of [99] had similar considerations when dealing with this issue, but they used SVM as a classification algorithm. They explained that they were more convinced than previous studies of the importance of selecting predictors so that reliable probabilistic RI predictions could be given. An ensemble of AI methods used for quantifying the RI predictability was presented in [100], which improved the skill 30% compared with the climatology model.

Another study illustrated how K-means can be used to analyze the favorable environment of the upper-tropospheric trough. Fischer et al. [101] examined the upper tropospheric potential vorticity structure, the TC convective structure and the TC environment through a composite analysis of rapidly intensifying TCs and non-rapidly intensifying (non-RI) TCs, which resulted in rules that can accelerate the RI.

### 3.4. TC Weather and the Disastrous Impact Forecasts

The primary reason why TC weather is defined as an extreme weather event is due to the fact that TC-induced meteorological and oceanic phenomena, such as high winds, heavy rains, and storm surges, are extremely destructive and devastating to humans [103]. If these techniques for forecasting disastrous weather and impacts are highly accurate, they will provide important guidance on taking protective measures in TC affected areas to minimize the loss of life and property [104–106]. However, TC wind fields, heavy rainfall, and storm surges involve complex interactions with the atmosphere and ocean at different scales; thus, they are difficult to predict. Especially with the forecasts of intense precipitation caused by TCs, the skill is still very limited for both short-term and long-term forecasts, although some efforts have been made [107–109].

According to previous studies, the accuracy of TC weather and disastrous impact forecasting depends on the development of observation technology and the accuracy of path and intensity forecasts [110,111]. As a result, their forecasting remains relatively poor compared to track forecasts. Nevertheless, there are many successful applications of machine learning in wind, rain, and tide forecasting. Scientists also believe that high winds, heavy rains, and storm surges during TC motion can be effectively predicted by machine learning, and this idea has been shown to not only improve existing forecasting models but also give more reliable instructions for hazard warnings (see Table 4).

### 3.4.1. TC Wind Field Forecasts

There is no doubt that a TC, once generated, can cause direct damage to offshore and coastal areas near the sea. Before landfall, strong winds blow through the sea, generating huge energy affecting the ocean's subsurface and generating waves that erode the beach. After making landfall, it will strike with significant destructive power and speed at urban roads and traffic, buildings and houses, farmland and crops, etc. [112].

In order to predict the hourly surface wind speed over the nearshore islands during TCs, Chih-Chiang et al. [113] used a kernel-based SVR model consisting of four kernel functions: radial basis, linear, polynomial, and Pearson VII kernel. The results showed that Pearson VII SVR was able predict the wind speed during a TC from 1 to 6 h, and the prediction was more effective than SVR models with other kernels, regression models, and parameterized representations of wind speed.

In turn, the feasibility of this machine learning-based technique in predicting sea surface wind speeds over offshore islands was demonstrated. Except from this study predicting the wind speed directly, other studies focused on the simulation of TC wind field using machine learning.

**Table 4.** Machine learning in TC weather and the disastrous impact forecasts.

| Tasks | Algorithms | Main Idea | Reference |
|---|---|---|---|
| TC wind field forecasts | SVR | Develop a highly reliable surface wind speed prediction technique | Wei, C.C. (2015). [113] |
| | LSSVM | Estimate TC innercore 2D surface wind field structure | Zhang, C. (2017). [114] |
| | MLP | Simulate the wind field inside the TC boundary layer | Snaiki, R. (2019). [115] |
| | | Construct a wind velocity simulation model | Wei, C.C. (2017). [116] |
| | | Optimize TC winds from satellite data | Stiles, B.W. (2014). [117] |
| TC rainfall forecasts | SVM | Design SVM-based models for the forecasts of hourly TC rainfall | Lin, G.F. (2009, 2013). [118,119] |
| | | Design a two-stage TC-induced flood forecasting model | Lin, G.F. (2013). [120] |
| | MLP | Develop a shallow MLP to forecast TC rainfall | Lin, G.F. (2005). [121] |
| | SOM, MLP | Develop a hybrid neural network model to forecast TC rainfall | Lin, G.F. (2009). [122] |
| | ANN-MRA | Build a model for forecasting the total rainfall and the groundwater level | Hsieh, P.C. (2019). [123] |
| | FNN-LLE | Design a TC precipitation prediction scheme | Huang, Y. (2018). [124] |
| Storm surge forecasts | MLP | Predict the short-term surge and surge deviation | Lee, T.L. (2007, 2009). [125,126] |
| | SVR | | Rajasekaran, S. (2008). [127] |
| | BPN-ANFIS | Quantify the RI predictability using an ensemble of AI methods | Chen, W.B. (2012). [128] |
| | MLP | Develop a time-dependent storm surge model for quick prediction | Kim, S.W. (2015). [129] |
| | | Predict the peak values of storm surge using the tropical storm parameters | Lewis, M.R.H. (2016). [130] |
| | ANN-SFM | Descibe a storm surge forecast model and an objective selection procedure | Kim, S. (2016, 2019). [131,132] |

In [114], the authors developed a machine learning method for estimating the two-dimensional (2D) surface wind field structure of the TC inner-core using infrared satellite images. Their main idea was to introduce least squares support vector machine (LSSVM), RBF, and linear regression to build two models. This is to simulate the relationship between the eye size and the maximum wind radius, and to derive the inner-core wind structure from the relevant information of the TC itself and the wind speed. The results showed that the wind field structure estimated by LSSVM was significantly better than the RBF and linear regression methods. To simulate the wind field inside the boundary layer of TCs, a knowledge-enhanced deep learning algorithm was proposed in [115]. More specifically, the regularization mechanism was enhanced by combining machine-readable knowledge of physical equations and/or semi-empirical formulas during the training of deep networks for TC boundary layer winds.

There is also a study that used the DL algorithm to determine the wind speed during numerical simulation [116]. They built a wind speed simulation model based on a MLP network and used the WRF model as a numerical model to computationally solve for wind speed at any location where the wind direction cannot be measured. The results showed that the proposed MLP combined with the WRF model could be effectively used to simulate the wind speed at any location in the study area. In addition, the authors of [117] developed a neural network technique for optimizing the wind

inversion from Ku-band scatterometer measurements in tropical cyclone conditions. When compared with wind speed data from aircraft observations, the new product using a neural network revealed 1–2 m/s of positive overall bias and a 3 m/s MAE.

### 3.4.2. TC Rainfall Forecasts

TC rainfall is different from normal rainfall events and requires consideration of not only the physics mechanisms of rainfall but also the impact of TCs [133,134]. Since about 2005, several researchers have used machine learning to predict TC rainfall. The first and main contributor was Lin and his team, who, in 2005, developed a neural network with two hidden layers to predict rainfall, using TC features and spatial rainfall information as input to the model. This model was proven to produce reasonable predictions; however, the model could only give accurate forecasts 1–2 h in advance [121].

In order to further improve the forecast techniques with a long lead-time and obtain more efficient results of the hourly rainfall forecasts, a novel SVM-based model was proposed by [118], with hourly rainfall characteristics as key inputs to the model. They recommended that the proposed SVM-based model as an alternative to the existing model at that time. Since heavy rainfalls can cause flooding, their team went on to present an SVM-based model of TC precipitation-induced flood forecasting [120]. In the model, the first step was to use the observed TC features and rainfall into the rainfall forecast; the second step was to use the predicted rainfall and observed runoff into the runoff forecast. The results showed that the SVM model generated accurate rainfall and runoff forecasts with a lead time of 1–6 h, especially for the peak runoff values, and the effectiveness of flood forecasting in the forecast time of 4 to 6 h was substantially improved compared to previous models.

Considering the limitations of a single algorithm, Lin et al. [122] also proposed a hybrid neural network model consisting of SOM and MLP, which could discover topological relationships during TC rainfall. In this model, SOM was used for clustering and discriminant analysis, while MLP was used to build the relationship between the input and output. The results showed that the model was more accurate in predicting the TC rainfall than the model constructed by the traditional neural network approach.

A similar study [119] developed a new hybrid TC rainfall forecasting model to improve hourly TC rainfall forecasts, integrating a multi-objective genetic algorithm with SVM. The advantage of this hybrid model is that meteorological parameters other than rainfall data are considered and this allows a choice of the optimal combination of input variables for each lead time. Another predictive model that uses MLP and multiple regression analysis (MRA) to predict total rainfall and groundwater level changes was shown to be able to improve the average accuracy of the model to 80% even when the training data were insufficient [123].

In addition, the authors of [124] developed a TC precipitation prediction model based on a fuzzy neural network (FNN) using National Center for Environmental Prediction (NCEP)/National Center for Atmospheric Research (NCAR) reanalysis data as potential predictors. Stepwise regression (SRM) and local linear embedding (LLE) algorithms were used to construct the input of a FNN model. The results showed that the RMSE values of the FNN-LLE model were 21.94, 24.07, and 25.22 for the TC precipitation predictions, which was better than the interpolation of ECMWF and SRM.

### 3.4.3. Storm Surge Forecasts

A storm surge caused by the strong wind of TCs can raise the coastal water level by 5–6 m, especially when the storm surge encounters astronomical high tides, which can produce high frequency tide levels and lead to seawall breakage and indwelling [135]. Therefore, it is important to predict storm surge changes nearshore during TCs. Traditional methods of forecasting storm surges are mainly numerical hydraulic models or empirical formulas [111], while machine learning has great potential to improve the statistical relationships in empirical formulas, which can give more accurate and faster forecasting results.

Since 2007, a study had been conducted using MLP to predict short-term TC surges and surge deviation to overcome the problem that nonlinear relationships have not been considered [125]. By comparing with the numerical methods, they found that the MLP model could effectively predict short-term storm surges 1 to 6 h in advance. Subsequently, the authors improved the MLP model by selecting wind speed, wind direction, air pressure, and astronomical tide level as inputs to the neural network, and validated the performance of the improved MLP model using the observational data obtained during the TCs [126].

During this period, SVR was also used as an emerging AI tool for storm surge forecasting [127]. Comparisons with numerical methods and neural networks showed that the SVM model was equally effective in predicting storm surges and surge deviations. Chen et al. [128] explored a hybrid neural network with back-propagation neural network (BPN) and an adaptive neural fuzzy inference system (ANFIS) algorithm to predict the storm surge height using the computational difference corrected by a two-dimensional hydrodynamic model, and the ANFIS model predicted the astronomical tide height and storm surge height well, with the lowest MAE and RMSE compared to the hydrodynamic model and BP model at different sites.

In addition, there are many other MLP-based models that also produce satisfactory results. For example, the authors of [129] produced a time-dependent surrogate model of storm surges based on MLP with synthetic simulations of TCs, and this model performed well on the nearshore. The authors of [130] developed an alternative and robust MLP-based model for predicting the peak values of storm surges, and they made available an unprecedented dataset that can be used to train AI models for storm surge prediction in these regions. In [131], the authors demonstrated an approach to generate more reliable surge forecasts models by selecting an MLP with a suitable dataset. They determined that the combination of the dataset of surge level, sea-level pressure, drop of sea-level pressure, longitude and latitude of TCs, sea surface level, wind speed, and wind direction were optimal for predicting the surge level, combined with an optimal lead time of 24 h.

The last typical case is shown in [132], where the authors described a novel approach with a systematic and objective selection procedure for an artificial neural network-based storm surge forecasting model (ANN-SFM), which has the advantage of being applicable to the development of storm surge forecasting models for other coastal areas, as it can combine the most relevant number of units and optimal input parameters.

### 3.5. Improving Numerical Forecast Models

Numerical weather forecasting (NWP) is based on the dynamical equations of the atmosphere and ocean to predict the weather based on the current weather conditions, which also includes TC forecasts. There are many different types of TC forecasting models in early warning centers around the world, for example, regional models—the Hurricane Weather and Research Forecast Models (HWRF); global models—the European Medium Range Weather Forecast Center's Global Model (EMX); and, aggregate models—the Florida Super Aggregate (FSSE) [136]. In general, the accuracy of TC numerical forecast models for genesis and intensity prediction is relatively low, mainly due to the inaccurate vortex initialization of TCs, incomplete representation of complex physical processes, and coarse resolution of the models [2,3]. The large datasets and complex calculations required for modern NWP must be accomplished with the help of the world's most powerful supercomputers, which are too costly. In addition, the density and quality of the observed data used as inputs for forecasts, as well as the defects in the numerical model itself, affect the accuracy of numerical forecasts. Although post-processing techniques, such as model output statistics (MOSs), have been developed to reduce errors in numerical predictions, this is far from the forecast accuracy required [137].

As far as we know, machine learning has been shown to provide a new approach to improving the parameterization of traditional NWP models. For example, researchers have made success in optimizing the planetary boundary layer, convection, clouds and ocean eddy parameterization, etc. using deep neural networks [138–143]. Therefore, in order to better produce accurate and fast TC

prediction results, machine learning is also used to improve existing numerical TC forecast models. According to the recent literature, cases involving machine learning to improve TC numerical models can be divided into three categories: pre-processing, the model itself, and post-processing (see Table 5). Preprocessing means to optimize the data assimilation methods so that the initial conditions of the model can be represented more accurately. For the model itself, the improvement lies in the error correction and parameterization of the model. Post-processing can be considered as better use of the numerical prediction products for TC forecasts.

**Table 5.** Machine learning in improving numerical forecast models.

| Tasks | Algorithms | Main Idea | Reference |
|---|---|---|---|
| Pre-processing | SVM | Produce the probability distribution of the TC regions in the data to be assimilated | Lee, Y.J. (2019). [144] |
| Improve model | RF, PCA | Replace simple TC winds parametric formulations with ML alogrithms | Loridan, T. (2017). [145] |
| | MLP, CNN | Parameterize SSTC induced by TCs to improve TC numerial models | Wei, J. (2017, 2018). [146,147] |
| Post-processing | CNN | Predict TC genesis using simulated OLR data from NICAM | Matsuoka, D. (2018). [46] |
| | | Dectect TCs in the outputs dataset from climate models | Liu, Y. (2016). [148] Racah, E. (2017). [149] Kim, S. (2017). [150] |
| | ConvLSTM | Predict TC paths based on large-scale data generated by climate models | Kim, S. (2019). [62] |
| | DT | Mine predictors of TC recurvature and landfall using the GFS-FNL dataset | Zhang, W. (2013). [64,65] |
| | SVM, MLP, RF | Predict RI using the output dataset of GFS | Mercer, A. (2017). [100] |

### 3.5.1. Pre-Processing

Data assimilation is a basic and classical method of giving the model an ideal initial field. It is difficult to replace the assimilation equation with machine learning, and replacing the cost function with machine learning is not yet possible. The relevant research is to improve the quality of the data before placing it into a data assimilation system and then to improve the forecasts results more effectively. One method to improve data assimilation is to identify specific regions in the datasets before entering the assimilation system, such as TC regions. By performing specific operations on these specific areas, the NWP prediction results can be improved more effectively. For example, the authors of [144] attempted to train a model to detect TC regions in satellite precipitation data using GFS model data and TC best track datasets. The model first redefined the TC region and then applied the SVM to classify the TC region and the no-TC region, finally giving the probability distribution of the TC regions. However, while this study offered a new approach to data processing, there remains room for improvement in the predicted results.

### 3.5.2. Improved Models

Machine learning methods may serve as tools to effectively simulate vortex initialization, reduce parameterization errors, and improve resolution, thus attracting researchers to explore them in depth. Previously, modeling wind uncertainty was a challenging topic, and traditional empirical equations have difficulty in describing the real wind field associated with TCs. In [145], PCA and RF were used to train the model to predict the conditional distribution of the first three principal component weights, thus providing a method to model the uncertainty of the wind field. This study may be able to provide a viable way of thinking for more precise representation of the initial vortex structure in the future.

There are also several typical cases in improving the parameterization scheme in NWP models. Wei and Jiang et al. [146,147] defined SSTC induced by TC motion using neural networks and analyzed its effect on TC numerical forecasts. Experiments showed that WRF considering MLP-defined SSTC had a great improvement on intensity forecasting, while deep neural networks (i.e., CNN)-defined SSTC in WRF had better predictions than MLP, close to the coupled model. They strongly indicated that the use of machine learning to improve the parameterization scheme of numerical models holds great promise in exploring more advanced TC numerical forecast techniques.

### 3.5.3. Post-Processing

For the post-processing of NWP model products for TC forecasting, there are two main methods in which machine learning can be used. One is to correct errors of the predictions of TC numerical models, and the other is to apply model products as the training and input data of machine learning models, and then to provide the forecast results. However, no machine learning-based error correction methods for TC numerical models have been found yet. There have been several successful cases for TC genesis, tracking, and intensity forecasts by using the outputs of numerical models.

For example, Matsuoka et al. [46] proposed a CNN-based approach based on twenty years of simulated outgoing longwave radiation (OLR) calculated using a Cloud-Resolving Global Atmospheric Simulation (NICAM, Nonhydrostatic Icosahedral Atmospheric Model) for identifying TCs and their precursors. Other studies demonstrated similar ideas to [46], utilizing deep CNNs to detect TCs in the output datasets of climate models [148–150]. The authors of [148] showed that their deep CNN system achieved 89~99% accuracy in detecting extreme events, like TCs, and the authors of [149,150] used semi-supervised learning algorithms and super resolution reconstruction network for improving the results.

In track forecasts, Kim et al. [62] designed ConvLSTM-based spatio-temporal models to track and predict hurricane trajectories from large-scale data in climate models, and Zhang et al. [64,65] used higher resolution and the time-period availability of meteorological variables from the National Centers for Environmental Protection (NCEP)-Global Forecasting System (GFS) Final Analysis (FNL) dataset for mining predictors of TC recurvature and landfall. There is also a study, [100], that shows an ensemble of machine learning methods using the GFS-reforecast (hereafter GFSR) dataset for TC rapid intensification.

## 4. Discussion

Machine learning has achieved progress in different aspects, including TC genesis, tracking, intensity, TC weather, and impact forecasts; however, many difficult and unsolvable problems remain in this field. This new way of thinking and new method of coping with key issues are both opportunities and challenges for TC forecasters and researchers.

### 4.1. Opportunities

(1) At present, the large-scale satellite data, reanalysis data, and observations data have not been fully developed and utilized, and machine learning has proven able to effectively detect and study various problems in remote sensing. Therefore, the prediction of TCs based on multi-source data, especially real-time data from satellites, is a promising topic.

(2) There are many bottlenecks in TC forecasts, such as the quantitative forecasts of cyclogenesis, the prediction of TCs with anomalous paths or rapid intensification, the intense precipitation caused by TCs, and TC wind field forecasts. Although there have been some preliminary attempts, they could not yet meet the requirements of operational forecasts, and these require further exploration in the future.

(3) It cannot be denied that numerical forecasts are still the dominant means of TC forecasting at this stage, and their importance should not be ignored. To improve the performance of numerical models in TC forecasts, machine learning could be used for an integration with numerical models,

including improving the parameterization scheme, replacing the sub-models represented by empirical formulas, or revising the deviation of outputs of the numerical models.

(4) Although numerical models are currently irreplaceable, they are expensive to run and have many physical processes that cannot be expressed by equations. Error propagation during model solving is also a big problem that contributes to poor forecasting. Therefore, the development of a pure data-driven TC prediction system that ensures high efficiency and low cost, while providing more accurate and stable TC predictions, is also the focus of future research by researchers.

*4.2. Challenges*

(1) The inexplicability of machine learning, especially deep learning, has been discredited by many experts in the meteorological field for its ability to predict TCs in a realistic and stable manner, because they consider that machine learning only discovers rules hidden within the data, and is detached from the real physical rules. How to make machine learning-based predictive models more stable and the prediction process more convincing is a new challenge.

(2) Machine learning in existing studies typically only provides short-lead-time predictions, or the accuracy does not meet expectations when making long-lead-time predictions. How to make longer and more accurate predictions is a bottleneck that needs to be overcome in the near future.

(3) A TC is a rare but extremely fast-changing and complex system with a large part of its lifetime over oceans, so for TCs, in-situ observations are scarce. Although there are a great deal of reanalysis data and satellite observations, as well as some airbone reconnaissance data available to study TCs, this is far from being enough to help us understand how real TCs happen and change. Therefore, to prove whether machine learning can capture the internal rules inside TCs based on limited data to make reliable predictions will still require extensive experiments and observations.

(4) The majority of machine learning-based TC predictive models at this stage are supervised learning methods, and TCs, an extreme weather phenomenon that cannot be quantitatively described in the real world, cannot be used directly as labels. How to reasonably construct labeled data and training datasets to train machine learning models to achieve predictive goals is also a question worth pondering. How unsupervised and semi-supervised learning methods should be effectively used in TC forecasting also requires further research.

## 5. Conclusions

Tropical cyclones have been a concern of meteorologists for more than 100 years. Numerous scholars have conducted in-depth studies on key issues, such as the structure, dynamics, and forecasting techniques. Machine learning is derived from statistical methods that can automatically discover relevant rules from large-scale data for detection, analysis, prediction, etc. The application of machine learning for the key problems of TCs provides a new way of thinking to address many bottlenecks in this field. Techniques based on a pure data-driven approach and using machine learning to improve numerical models have both been shown by a large number of studies to provide huge contributions to improving TC predictions. Although existing research has made some progress in genesis forecasts, path prediction, intensity prediction, TC weather prediction, and improving numerical forecast models by integrating machine learning, there are still many aspects that remain to be studied, which we regard as both an opportunity and a challenge.

The opportunity is that the potential of machine learning has not yet been exploited, and large-scale data are still underutilized. The challenge is that TCs are different from normal weather phenomena and oceanic physical processes in that they are complex, subject to many factors, and it is difficult to obtain comprehensive in situ observations inside TCs. We can conclude that machine learning in TC forecasts is both promising and challenging, which means that it requires researchers to have a good understanding of TC dynamics as well as a knowledge of machine learning in order to discover the key problems faced and to solve them by building suitable machine learning models. By analyzing and summarizing the studies of machine learning in TC forecasts over the years,

we hope this review can provide readers with insight into this research and lay a foundation for future works regarding machine learning in TC forecast modeling.

**Author Contributions:** R.C. contributed to investigation, collection and organization of literature, framework, original draft preparation, improvement and revision, and final manuscript. W.Z. contributed to theme selection, methodology, framework, supervision, and manuscript revision. X.W. contributed to framework, improvement, and manuscript revision. All authors have read and agreed to the published version of the manuscript.

**Funding:** This research was funded by the National Key Research and Development Program of China (Grant No. 2018YFC1406206, 2018YFC1406202) and the National Natural Science Foundation of China (Grant No. 61802424)

**Acknowledgments:** Most of the cases in this review are included in Web of Science, so we are very grateful for the Foreign Language Database containing the Web of Science provided by the Library of the National University of Defense Technology.

**Conflicts of Interest:** The authors declare no conflict of interest.

## Abbreviations

The following abbreviations are used in this manuscript:

| | |
|---|---|
| TC | Tropical cyclone |
| ML | Machine learning |
| AI | Artificial Intelligence |
| SVM | Support vector machine |
| DT | Decision tree |
| ANN | Artificial neural network |
| CNN | Convolutional neural network |
| RNN | Recurrent neural network |
| DL | Deep learning |
| SAE | Sparse autoencoder |
| SAR | Synthetic aperture radar |
| ConvLSTM | Convolutional long short-term memory network |
| TrajGRU | Trajectory gate recurrent unit |
| SVR | Support vector regression |
| RF | Random forest |
| BPN | Back-propagation neural network |
| MLP | Multi-layer perceptron |
| LSTM | Long short-term memory neural network |
| CISK | Convective instability of the second kind |
| WHISE | Wind-induced surface heat exchange |
| ECMWF | European Centre for Medium-Range Weather Forecasts |
| GFS | Global Forecast System |
| UKMET | United Kingdom Meteorological Forecast Model |
| LDA | Linear discriminative analysis |
| NB | Naïve bayes |
| KNN | K-nearest neighbor |
| QDA | Qualitative data analysis |
| PCA | Principal component analysis |
| LR | Logistic regression |
| MCS | Mesoscale convective system |
| NOGAPS | Navy Operational Global Atmospheric Prediction System |
| OLR | Outgoing longwave radiation |
| QBO | Quasi-Biennial Oscillation |

| | |
|---|---|
| MAE | Mean absolute error |
| RMSE | Root mean square error |
| GPI | Genesis potential index |
| SOM | Self-organizing map |
| NOEGM | Neural oscillatory elastic graph matching model |
| HRBF | Hybrid radial basis function network |
| TDSL | Time difference and structural learning |
| OTCM | One-way-interactive Tropical Cyclone Model |
| JTWC | Joint Typhoon Warning Center |
| TFS | Track Forecast System |
| CLIPER | Climatology and Persistence |
| LAM | Florida State University based Limited Area Model |
| QLM | Quasi Lagrangian Model |
| NHC | National Hurricane Center |
| MNN | matrix neural network |
| GRU | Gated Recurrent Unit |
| ENSO | El Niño-Southern Oscillation |
| MJO | Madden-Julian Oscillation |
| CART | Classification and regression tree |
| GAN | Generative adversarial networks |
| DBN | Deep belief networks |
| SSM/I | Special Sensor Microwave/Image |
| MLR | Multiple logistic regression |
| 2D-CNN | Two-dimensional convolutional neural network |
| 3D-CNN | Three-dimensional convolutional neural network |
| WNP | West Northern Pacific |
| JMA-GSM | Japan Meteorological Agency-Global Spectral Model |
| CNN-LSTM | Convolutional neural network-Long short-term memory neural network |
| WP | West Pacific |
| EA | Evoutionary algorithm |
| PSO | Partical Swarm Optimization |
| OC_PI | ocean coupling potential intensity index |
| RI | Rapid intensification |
| non-RI | No rapidly intensifying |
| LSSVM | Least squares support vector machine |
| WRF | Weather Research and Forecasting |
| MRA | Multiple regression analysis |
| NCEP | National Centers for Environmental Prediction |
| NCAR | National Center for Atmospheric Research |
| FNN | Fuzzy neural network |
| SRM | Stepwise regression method |
| LLE | Locally linear embedding |
| FNN-LLE | Fuzzy neural network-Locally linear embedding |
| ANFIS | Adaptive neuro-fuzzy inference system |
| NWP | Numerical weather prediction |
| ANN-SFM | Artificial neural network-based storm surge forecast model |
| HWRF | Hurricane Weather and Research Forecasting Model |
| EMX | European Center for Medium-Range Weather Forecasts Global Model |
| FSSE | Florida State Super Ensemble |
| MOS | Model output statistics |
| SSTC | Sea surface temperature cooling |
| NICAM | Nonhydrostatic Icosahedral Atmospheric Model |
| GFS-FNL | Global Forecasting System-Final Analysis Dataset |

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
