# Peer review of "Machine Learning in Tropical Cyclone Forecast Modeling: A Review"

_atmosphere, doi:10.3390/atmos11070676_

Round 1

Reviewer 1 Report

  1. The paper should be carefully edited; English should be checked and corrected.
  2. Lines 51 - 52: ‘Artificial neural network (ANN) is also a classical algorithm to solve this kind of problems all the time.’ ANN is a universal approximator for complex nonlinear mappings. See:  Hornik, K., Stinchcombe, M., White, H. (1990). Universal approximation of an unknown mapping and its derivatives using multilayer feedforward network. Neural Networks, 3, 551-560; Hornik, K. (1991). Approximation Capabilities of Multilayer Feedforward Network. Neural Networks, 4, 251-257

Majority of works done with so called “regression algorithms” (especially when you have more than one output) were performed with ANN, not with SVM.  When you have several outputs, the problem becomes a mapping (relationship between two vectors) not a regression.  Mapping cannot be adequately represented by several (as many as many outputs) regressions (e.g., see:  Krasnopolsky, V. (2013). The Application of Neural Networks in the Earth System Sciences. Neural Network Emulations for Complex Multidimensional Mappings. Atmospheric and Oceanic Science Library. (Vol. 46). Dordrecht, Heidelberg, New York, London: Springer. DOI 10.1007/978-94-007-6073-8)

  1. Lines 62 – 67: Remote sensing applications also include developments of 4) fast and accurate retrieval algorithms and 5) fast and accurate forward models (e.g. see also Krasnopolsky V. 2013)
  2. Line 105 and Fig. 1: Your f in f: X -> Y is not a function; it is a mapping (as  it is properly stated at line 117) because both X and Y may be vectors (sets of parameters). 
  3. Lines 117 – 130: description of ML procedure is a bit confusing. Actually, mapping f depends also on a vector of nonlinear (in general) parameters, w: y = f(x,w). Parameters w are obtained in the process of training, which for regression/mapping problem is an optimization of performance criterion (e.g., minimization of mean square error).  A ML algorithm itself may have additional parameters (meta-parameters) like the number of hidden neurons and learning rates for neural networks.  It is not clear does “tuning” (term used by authors) mean the selection of meta-parameters, or is it used as synonym for “training”?  It should be clarified.
  4. Line 163 and 167, Fig. 2. ANN is a generic name for NN algorithms. CNN, RNN, and DNN are three particular members of the ANN family of algorithms.
  5. Lines 203 and 204. “In addition, existing machine learning techniques are not yet mature enough to comprehensively improve numerical forecast models.” It is not clear how this statement is related to the previous ones.
  6. It is difficult to agree with the authors statement that “Real data of TCs is difficult to measure and obtain”. To the best of my knowledge there exist extensive archives of TCs track and intensity data as well as satellite and other observations.  There exist also archives of numerical data produced by different TC models.

Reviewer 2 Report

Comments on “Machine Learning in Tropical Cyclones Forecasts

Modeling: A Review”

There are many challenging questions regarding tropical cyclones. In addition to numerical modeling, machine learning and data mining are useful ways to supplement these models for understanding and forecasting tropical cyclones. This work is a timely review of the status of machine learning in investigating tropical cyclones forecasts. This review is overall well organized and written. I support the publication of this work and provide here some comments to be addressed by the authors, from the perspectives of writing and references.

Lines 44-48: Actually, a commonly used clustering algorithm for tropical cyclone tracks is the finite-mixed-model-based method. Please refer to previous studies here (e.g., Camargo et al. 2007a;b; Geng et al. 2016; Zhang et al. 2013b).

Lines 321-326: In terms of TC track forecasts, Wang et al. (2011) and Zhang et al. (2018) used Back Propogation(BP)-neural network for forecasting western Pacific tropical cyclone tracks.

Lines 366-380: It would be good to mention a work that focuses on landfalling TCs over China using data mining (Geng et al. 2016).

Lines 375: landfalling TCs

Lines 382-405: Actually, evolutionary algorithm and decision tree methods (e.g., CART or C4.5) have also been used to forecast TC intensity change (Gao et al. 2016; Geng et al. 2015; 2017; Zhang et al. 2013a). Some discussions would be helpful.

Line 400: This sentence is incomplete. Please use “need to address”.

Line 409: “algorithms is also”

Line 462: “improve TC forecasts”

Lines 459-478: Decision tree methods have been used to examine rapid intensity change (e.g., Gao et al. 2016; Zhang et al. 2013a).

Line 481: change “such as high winds, heavy rains, storm surges, etc.” to “such as high winds, heavy rains, and storm surges”

Line 497: change “Once landed” to “After making landfall”

Line 543: “Guofang Lin et al.[99]”? Please make sure this is correct.

Line 639: “under” to “associated with”. In addition, I am slightly struggling to consider “principal component analysis” as “machine learning techology”.

Line 644: “Jun Wei et al.”? Make sure this is correct. Which reference is this?

Line 674: remove “etc.”

Line 683: Could the authors add more discussions in terms of “using machine learning on studying intense precipitation associated with TCs”? Indeed, this is a very challenging aspect of TC study and machine learning may help us. While some efforts have been made to forecast intense precipitation caused by TCs (Gao et al. 2011; Lonfat et al. 2007; Marchok et al. 2007; Zhang et al. 2019), the skill is very limited for both short-term and long-term forecasts of this variable (i.e., TC rainfall).

Line 694: What is “display equations”?

Line 718: “various of” to “numerous”

Reference

  1. Camargo, S. J., A. W. Robertson, S. J. Gaffney, P. Smyth, and M. Ghil (2007a), Cluster Analysis of Typhoon Tracks. Part I: General Properties, Journal of Climate, 20(14), 3635-3653, doi:10.1175/JCLI4188.1.
  2. Camargo, S.J., A.W. Robertson, S.J. Gaffney, P. Smyth, and M. Ghil, 2007b: Cluster Analysis of Typhoon Tracks. Part II: Large-Scale Circulation and ENSO. Climate, 20, 3654–3676.
  3. Gao S., Zhang, W., Jia Liu, I.-I. Lin, Long S. Chiu, and Kai Cao, 2016, Improvement of typhoon intensity change classification by incorporating an ocean coupling potential intensity index in decision trees, Weather and Forecasting, 31, 95–106.
  4. Gao, S., and L. S. Chiu (2011), Development of Statistical Typhoon Intensity Prediction: Application to Satellite-Observed Surface Evaporation and Rain Rate (STIPER), Weather and Forecasting, 27(1), 240-250, doi:10.1175/WAF-D-11-00034.1.
  5. Geng, H., Shi, D., Zhang, W., Huang, C., 2016, A Prediction Scheme for the Frequency of Summer Tropical Cyclone Landfalling over China based on Data Mining Methods, Meteorological Applications, 23, 587–593
  6. Geng H., J. Sun, W. Zhang and C. Huang, "Study on Index Model of Tropical Cyclone Intensity Change Based on Projection Pursuit and Evolution Strategy," 2015 IEEE Symposium Series on Computational Intelligence, Cape Town, 2015, pp. 145-150, doi: 10.1109/SSCI.2015.31.
  7. Geng, Huan-tong, S. Jia-qing, Z. Wei, and W. Zheng-xue (2017), A NOVEL CLASSIFICATION METHOD FOR TROPICAL CYCLONE INTENSITY CHANGE ANALYSIS BASED ON HIERARCHICAL PARTICLE SWARM OPTIMIZATION ALGORITHM, Journal of Tropical Meteorology, 23(1).
  8. Lonfat, M., R. Rogers, T. Marchok, and F. D. Marks Jr (2007), A parametric model for predicting hurricane rainfall, Monthly Weather Review, 135(9), 3086-3097.
  9. Marchok, T., R. Rogers, and R. Tuleya (2007), Validation schemes for tropical cyclone quantitative precipitation forecasts: Evaluation of operational models for US landfalling cases, Weather and forecasting, 22(4), 726-746.
  10. Wang Y., W. Zhang and W. Fu, "Back Propogation(BP)-neural network for tropical cyclone track forecast," 2011 19th International Conference on Geoinformatics, Shanghai, 2011, pp. 1-4, doi: 10.1109/GeoInformatics.2011.5981095.
  11. Zhang, W., Gao, S., Chen, B., Cao, K., (2013a), The application of decision tree to intensity change classification of tropical cyclones in the western North Pacific, Geophysical Research Letters, 40, 1883-1887.
  12. Zhang, W., Leung, Y., Wang, Y. (2013b): Cluster Analysis of Post-landfall Tracks of Landfalling Tropical Cyclones over China, Climate Dynamics,5-6, 1237-1255.
  13. Zhang, W., Villarini, G., Vecchi, G.A. et al. Rainfall from tropical cyclones: high-resolution simulations and seasonal forecasts. Clim Dyn 52, 5269–5289 (2019).
  14. Zhang Y., R. Chandra and J. Gao, "Cyclone Track Prediction with Matrix Neural Networks," 2018 International Joint Conference on Neural Networks (IJCNN), Rio de Janeiro, 2018, pp. 1-8, doi: 10.1109/IJCNN.2018.8489077.

Reviewer 3 Report

(General comments)

In this review, studies related to machine learning application for Tropical cyclone forecasting are summarized.

The reviewer felt that the manuscript is well structured from broad point of view, and it could be useful for readers.

On the other hand, it is better to revise or correct:

  • some English expressions
  • explanations of abbreviations (give some description in first appearances. e.g. GAN, DBN, RI, etc)
  • styles of references

The reviewer recommends to apply professional English copyediting service (by native English speakers) before submission of the revised manuscript.

(Minor comments)

  • L.18: Tntensity -> Intensity?
  • L.21: regard -> regarded
  • L.84-85: If possible, it is better to include unsuccessful cases.
  • L. L.134: et al. -> "and so on" or "etc"
  • L.186-187: The reviewer thinks that not only "probabilistic forecasts" is final goal, but quantitative forecast (e.g. time and place of cyclogenesis) is also the purpose of TC genesis forecast.
  • L.169-170: What is "unexpected result"? Good? or bad?
  • L.158: Navy ... Forecast System -> Navy ... Prediction System
  • L.330: What is "the bureau numerical TC forecast model"?
  • L.384: known -> know
  • L.372: What is "C4.5"?
  • L.528: serval? typo?
  • L.682-687: Not so important paragraph. It can be omissible.
